# Convergent Evolution of Himalayan Marmot with Some High-Altitude Animals through ND3 Protein

**DOI:** 10.3390/ani11020251

**Published:** 2021-01-20

**Authors:** Ziqiang Bao, Cheng Li, Cheng Guo, Zuofu Xiang

**Affiliations:** College of Life Science and Technology, Central South University of Forestry and Technology, Changsha 410004, China; bao_ziqiang@163.com (Z.B.); lslicheng@126.com (C.L.)

**Keywords:** convergent evolution, phylogenetic tree, genetic distances, *Marmota himalayana*, ND3

## Abstract

**Simple Summary:**

The Himalayan marmot (*Marmota himalayana*) lives on the Qinghai-Tibet Plateau and may display plateau-adapted traits similar to other high-altitude species according to the principle of convergent evolution. We assessed 20 species (marmot group (n = 11), plateau group (n = 8), and Himalayan marmot), and analyzed their sequence of *CYTB* gene, CYTB protein, and ND3 protein. We found that the ND3 protein of Himalayan marmot plays an important role in adaptation to life on the plateau and would show a history of convergent evolution with other high-altitude animals at the molecular level.

**Abstract:**

The Himalayan marmot (*Marmota himalayana*) mainly lives on the Qinghai-Tibet Plateau and it adopts multiple strategies to adapt to high-altitude environments. According to the principle of convergent evolution as expressed in genes and traits, the Himalayan marmot might display similar changes to other local species at the molecular level. In this study, we obtained high-quality sequences of the *CYTB* gene, CYTB protein, *ND3* gene, and ND3 protein of representative species (n = 20) from NCBI, and divided them into the marmot group (n = 11), the plateau group (n = 8), and the Himalayan marmot (n = 1). To explore whether plateau species have convergent evolution on the microscale level, we built a phylogenetic tree, calculated genetic distance, and analyzed the conservation and space structure of Himalayan marmot ND3 protein. The marmot group and Himalayan marmots were in the same branch of the phylogenetic tree for the *CYTB* gene and CYTB protein, and mean genetic distance was 0.106 and 0.055, respectively, which was significantly lower than the plateau group. However, the plateau group and the Himalayan marmot were in the same branch of the phylogenetic tree, and the genetic distance was only 10% of the marmot group for the ND3 protein, except *Marmota flaviventris*. In addition, some sites of the ND3 amino acid sequence of Himalayan marmots were conserved from the plateau group, but not the marmot group. This could lead to different structures and functional diversifications. These findings indicate that Himalayan marmots have adapted to the plateau environment partly through convergent evolution of the ND3 protein with other plateau animals, however, this protein is not the only strategy to adapt to high altitudes, as there may have other methods to adapt to this environment.

## 1. Introduction

Convergent evolution describes a phenomenon where different species independently develop similar morphology, physiology, and behavior when they are living in similar environments or under similar selective pressures [1,2]. For example, the yak (*Bos mutus*) and Tibetan mastiff are both resistant to altitude sickness because they live on high plateaus [3,4]. Since phenotypic traits are determined and regulated by genetic changes, genes will also change to reflect environmental adaptation [2], meaning that microscale changes eventually lead to ecological and morphological convergence.

Himalayan marmots (*Marmota himalayana*) live between altitudes of 2900 and 5500 m above sea level on the Qinghai-Tibet Plateau. This region is perennially cold, with low atmospheric oxygen levels and constant exposure to strong ultraviolet radiation [5,6]. Adaptation to this extreme environment was facilitated by a series of genetic mutations in the Himalayan marmot which produced advantageous physiological changes. Researchers have found that the expression of the genes *Slc25a14* and *ψ Aamp* differed when compared with Himalayan marmots living at lower altitudes through genome analysis [7]. If living at high altitudes drives convergent evolution, similar changes might be seen in other local animals.

NADH (nicotinamide adenine dinucleotide hydride) dehydrogenase is an important enzyme located on the inner membranes of mitochondria. It is involved in the NADH oxidation respiration chain, mediating cyclic electron transfer from NADH to coenzyme Q [8]. It plays a significant role in cell growth, differentiation, energy metabolism, and cell protection. ND3 (NADH dehydrogenase III), one of the NADH dehydrogenase subunits, also has a regulatory and protective effect in the oxidative respiration chain [9,10,11].

To explore whether plateau species have convergent evolution through the ND3 protein, the role of ND3 in energy metabolism and cellular respiration, and CYTB (cytochrome b), frequently used to resolve relationships between species [12,13], we constructed a phylogenetic tree for selected animal species based on those sequences and calculated genetic distances, paying particular attention to the ND3 protein in the Himalayan marmot. Twenty species were used in the study to represent the variables of living in a plateau environment (the plateau group) and relatedness to the focal marmot species (the marmot group), as well as including the Himalayan marmots themselves. The species forming the Qinghai-Tibet Plateau group included the yak (*Bos mutus*)*,* the plateau zokor (*Eospalax fontanierii baileyi*)*,* the kiang (*Equus kiang*)*,* the plateau pika (*Ochotona curzoniae*)*,* the snow leopard (*Panthera uncia*)*,* the Tibetan antelope (*Pantholops hodgsonii*)*,* the Yunnan snub-nosed monkey (*Rhinopithecus bieti*)*,* and the Sichuan snub-nosed monkey (*Rhinopithecus roxellana*). The marmot group included the gray marmot (*Marmota baibacina*)*,* the bobak marmot (*Marmota bobak*), the hoary marmot (*Marmota caligata*)*,* the black-capped marmot (*Marmota camtschatica*)*,* the long-tailed marmot (*Marmota caudata*)*,* the yellow-bellied marmot (*Marmota flaviventris*)*,* the alpine marmot (*Marmota marmota*)*,* Menzbier’s marmot (*Marmota menzbieri*)*,* the groundhog (*Marmota monax*)*,* the Olympic marmot (*Marmot olympus*), and the Tarbigan marmot (*Marmota sibirica*). The Himalayan marmot (*Marmota himalayana*) would be compared against both groups.

In this experiment, we hypothesized that the phylogenetic relationships of ND3 and CYTB would differ between the two groups in reference to Himalayan marmots, and that the ND3 protein of Himalayan marmots would show a history of convergent evolution at the molecular level.

## 2. Materials and Methods

### 2.1. Species Geographic Distribution

In this study, the geographic distribution and elevation of each species is shown in Figure 1A,C, and the twenty species under consideration were divided into the marmot group (n = 11) and the plateau group (n = 8), with the Himalayan marmot analyzed in a class by itself (Figure 1B). The members of the plateau group all live on the Qinghai-Tibet Plateau [14,15,16,17,18,19,20]. In contrast, the marmot group is geographically diverse. The gray marmot, bobak marmot, long-tailed marmot, and Menbzier’s marmot live in Central Asia [21,22,23,24,25]. The Tarbigan marmot lives across the Mongolian region [26,27]. The black-capped marmot and the alpine marmot live in Europe, while the hoary marmot, the yellow-bellied marmot, the Olympic marmot, and the groundhog are distributed across North America [28,29,30,31,32,33,34,35].

### 2.2. Sequence Database

All animal sequences of the *CYTB* gene, CYTB protein, *ND3* gene, and ND3 protein were obtained from https://www.ncbi.nlm.nih.gov/nuccore/ and https://www.ncbi.nlm.nih.gov/guide/proteins/ in the NCBI (National Center for Biotechnology Information) database. Except for certain unpublished *ND3* gene sequences, all others were downloaded as FASTA files from NCBI (Table 1).

### 2.3. Phylogenetic Analyses

We adopted two methods of phylogenetic analysis in this study. First, we constructed phylogenetic trees [36] of the *CYTB* genes, CYTB proteins, and ND3 proteins using software MEGA7 (Molecular Genetics Analysis Version 7) (Arizona State University, Arizona State, AZ, USA) via the Maximum Likelihood method (ML). These phylogenetic trees were used to visualize the relationship between the marmot group, the plateau group, and the Himalayan marmot. Species found along the same branch had greater similarity, and presumably closer relationships, while species on divergent branches were farther apart. Then, we calculated the genetic distance (pairwise distances) of the above sequences using the Kimura 2-parameter distance (K2P) model [37,38]. Genetic distance is an important measure of the similarity between genetic sequences: larger values indicate more divergent sequences and vice versa [39]. All genetic distances were calculated in comparison to the Himalayan marmot, which was set to a reference value of zero.

### 2.4. Consensus Sequence of ND3

After phylogenetic analysis was complete, we inferred the consensus amino acid sequence of the ND3 protein via Jalview Version 2.1 (University of Dundee, Scotland, UK) [40]. The ND3 protein sequences of all twenty study species were imported into Jalview in FASTA format. Consensus sites appeared deep blue, while uncertain sequences were marked with lighter colors. The consensus of all groups is summarized below. Nonconsensus sites are indicated with red rectangles, and special amino acid sequences in Himalayan marmots that match the plateau group, but not the marmot group, are circled.

### 2.5. Modelling ND3 Protein Tertiary Structure of Himalayan Marmot

The three-dimensional structure of the Himalayan marmot ND3 protein was predicted by homologous modeling as follows: First, the amino acid sequence (FASTA format) of the Himalayan marmot ND3 was imported into the Swiss-Model website http://swissmodel.expasy.org/ [41]. Second, we manually selected highly homologous protein sequences to use as target template sequences to construct the three-dimensional structure (tertiary structure) of the ND3 protein. Finally, we comprehensively evaluated the accuracy of the target protein structure using the qualitative model energy analysis (QMEAN) value, pattern identity, z-score, and Ramachandran plots.

### 2.6. Statistical Analysis

All significance analyses were calculated by means ± standard deviation, each datum of the group was set to three replicates. Statistical graph construction and one-way ANOVA were performed in Graphpad Prism 8 (GraphPad Software Inc., San Diego, CA, USA), and the significance threshold was set to *p*-value < 0.01. Graphpad Prism was also used to create heatmap plots of the *CYTB* gene, CYTB protein, and ND3 protein genetic distance, with red denoting high similarity.

## 3. Results

### 3.1. Relationship Analysis of CYTB Gene and CYTB Protein

CYTB is an important marker for classifying genetic relationships between species. Thus, we constructed phylogenetic trees of *CYTB* genes and CYTB proteins from all twenty species to verify evolutionary relationships. Phylogenetic tree branch-site analysis revealed that the Himalayan marmot occurred on the same branch as the marmot group, compared against the plateau group (Figure 2A and Figure 3A). As expected, we also found *Bos mutus* and *Pantholops hodgsonii* in the same lineage, as well as *Rhinopithecus bieti* and *Rhinopithecus roxellana*. When the relationship was quantitatively analyzed using genetic distance, the results showed the *CYTB* gene and CYTB protein genetic distances of the marmot group were 0.064–0.136 and 0.045–0.072, respectively (Figure 2B and Figure 3B, green bar), and for the plateau group the distances were 0.271–0.387 and 0.139–2.365 (Figure 2B and Figure 3B, blue bar), 3–10 times greater. Statistical analysis showed that the distances to the marmot group were significantly smaller than the distances to the plateau group for both the *CYTB* gene sequence and the CYTB protein sequence (*p* < 0.01) (Figure 2C and Figure 3C). The heatmaps also showed that the Himalayan marmots are closely related to all the other marmots (Figure 2D and Figure 3D).

### 3.2. Relationship Analysis of ND3 Proteins

Because the ND3 protein is related to the respiratory chain, we approached it as the target protein in this study. The plateau group and the Himalayan marmot are in the main branch of ND3 protein phylogenetic trees to the exclusion of the marmot group except for *Marmota flaviventris* (Figure 4A). When the genetic distance of the ND3 protein was quantitatively analyzed (Figure 4D), the distance for ND3 in reference to the Himalayan marmot was found to be between 0.232 and 0.397 for the plateau group and 2.115–3.367 for the marmot group, except for *Marmota flaviventris* (0.053). Statistical analysis showed that ND3 genetic distances for the plateau group were significantly lower than for the marmot group (*p* < 0.01) (Figure 4B). The heatmaps of the Himalayan marmot and the plateau group were much more similar to each other than to the marmot group (Figure 4E). To illustrate the connection between the variant ND3 protein and environmental conditions, we created a scatter diagram of the genetic distances for the CYTB protein and the ND3 protein (Figure 4C). The results revealed two distributions, with the plateau group having a positive correlation with the plateau environment and the marmot group tracking closely with CYTB.

### 3.3. ND3 Protein Sequence Analysis

To analyze the changes to the ND3 protein at the molecular level, we determined the conserved and nonconserved amino acid sequences of ND3 (the primary structure) (Figure 5A). The consensus of all species in the plateau group and the marmot group are reported in Figure 5B,C, respectively. The results show a number of conserved areas, but in the Himalayan marmot, the eighth amino acid in the sequence was L (leucine), similar to the plateau group, instead of the F (phenylalanine) found in that position in the marmot group. We infer that this site may be associated with convergent evolution in a high-altitude environment.

### 3.4. Modelling ND3 Protein Tertiary Structure of Himalayan Marmot

The properties of a protein are determined by its shape, so to find the shape of ND3, we used the homology modeling method to construct its three-dimensional structure (Figure 6A). The modeling result was that the target-template alignment protein, 5gpn.32.A (from the pig, *Sus scrofa*), was 84.52% similar to the ND3 protein. In the covered main region (Figure 6B,C), the Z-score > 2 (Figure 6D), and the QMEAN value was −2.94 (Figure 6B). The thumbs-up icon next to the QMEAN score indicates the target-template was close enough to be used. In the Ramachandran plots, more than 90% of the red dots (amino acid residues) were distributed in the allowed areas (dark green) and the maximum allowable areas (light green) (Figure 6E). This suggests that the dihedral-angles ψ and φ of the amino acid residues were in a reasonable area, conforming to the rules of stereochemistry.

## 4. Discussion

The study confirmed that the Himalayan marmot is more closely related to the marmot group than the plateau group according to the *CYTB* gene sequence and the CYTB amino acid sequence. In contrast, for ND3, which may be related to adapting to life at high altitudes, the sequences observed in the Himalayan marmot were closer to other animals from the Qinghai-Tibet Plateau, with genetic distances only 10% of that observed for the rest of the marmot group (except *Marmota flaviventris*). The scatter diagrams and heatmaps of ND3 and CYTB distances also reinforced these relationships. Further investigation revealed that the Himalayan marmot ND3 protein contained both conserved and nonconserved sequences and that the change in the eighth amino acid (from F to L) may change the protein’s tertiary structure. Taken together, these results suggest that the ND3 protein plays a vital role in adapting to high altitudes, and that it is an example of convergent evolution in animals living on the Qinghai-Tibet Plateau.

CYTB (cytochrome b) is the electron transfer in the mitochondrial membrane; the role of CYTB is to complete the reversal of the reduction state (Fe^2+^) and the oxidation state (Fe^3+^) of iron [42,43]. It is commonly used to measure the genetic relatedness between species. For instance, Guan et al. used CYTB as a marker to measure the relationships among goats (*Capra hircus*) [44]. In this study, we found that Himalayan marmots were significantly more closely grouped with other marmot species than any other animals living on the Qinghai-Tibet Plateau. The same methodology also grouped *Bos mutus* with *Pantholops hodgsonii* and *Rhinopithecus bieti* with *Rhinopithecus roxellana*, which is expected, given that the first two species belong to the family Bovidae [18] and the second two are both species of Chinese snub-nosed monkey [45]. This is consistent with the use of CYTB as a genetic marker to measure the relatedness of species.

To adapt to similar environmental stressors, animals may independently develop similar evolutionary strategies involving morphological, physiological, and behavioral changes—convergent evolution. On the Qinghai-Tibet Plateau, the metabolic role of the ND3 protein in the oxidative respiration chain could be relevant to animals like the Himalayan marmot adapting to low-oxygen conditions [8,9,10,11]. Accordingly, the ND3 sequences were more similar among the nine plateau animals. Interestingly, the yellow-bellied marmot also showed some of the changes in ND3 observed in the Himalayan marmot, with a genetic distance value lower than the plateau group (genetic distance = 0.052). The two species could live in similar environments, as the yellow-bellied marmot is known to live at high altitudes in the western United States and southwestern Canada [46,47], raising the possibility of parallel evolution.

To assess how the Himalayan marmot ND3 protein changed in response to the environment of the Qinghai-Tibet Plateau, we compared the amino acid sequences between the two groups and found many nonconserved sequences. The eighth amino acid in the Himalayan marmot ND3 sequence was leucine, similar to members of the plateau group, but not the marmot group (phenylalanine-typical). Phenylalanine is encoded by the codons UUU and UUC, whereas leucine is encoded as UUA, UUG, CUU, CUC, CUA, and CUG, meaning that there are several point mutations that can produce this change. A switch from phenylalanine to leucine will lead to changes in proteins from the primary structure to the tertiary structure. Through homologous modeling, we found that ND3 structure of Himalayan marmots was similar to the pig (*Sus scrofa*) protein sequence 5gPN.32.A (sequence similarity 84.52%), which may be related to the amount of energy required for domestic pigs’ muscle growth and fat metabolism [48]. This provides a theoretical basis for investigating another potentials function and effect of ND3.

In conclusion, Himalayan marmots are an example of convergent evolution in their adaptation to life on the Qinghai-Tibet Plateau. One such adaptation may be related to the ND3 protein, which has been restructured at the molecular level relative to other marmots.

## Figures and Tables

**Figure 1 animals-11-00251-f001:**
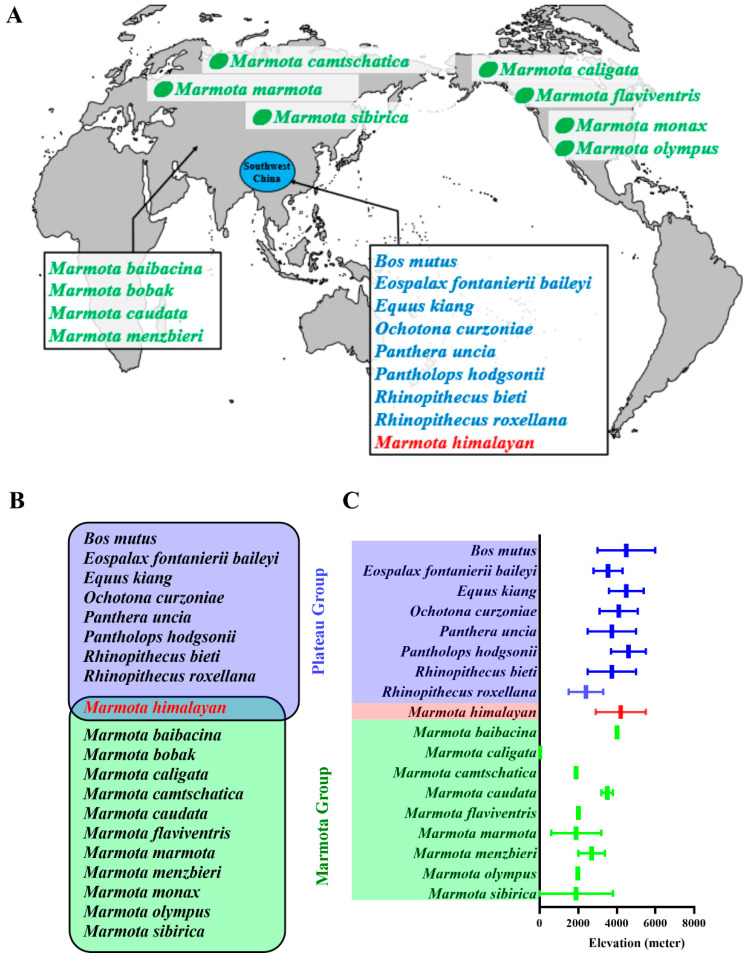
Relationship and geographic distribution of study species. The distribution of all study species is shown in map (**A**). The eleven members of the marmot group are highlighted in green, while the eight members of the plateau group are highlighted in blue. The focal Himalayan marmot is in red. The animals of the plateau group as well as the Himalayan marmot live on the Qinghai-Tibet Plateau (blue circle). Venn graph (**B**) shows this relationship, while graph (**C**) presents the elevations (with lower elevation limit and upper elevation limit) at which each species lives.

**Figure 2 animals-11-00251-f002:**
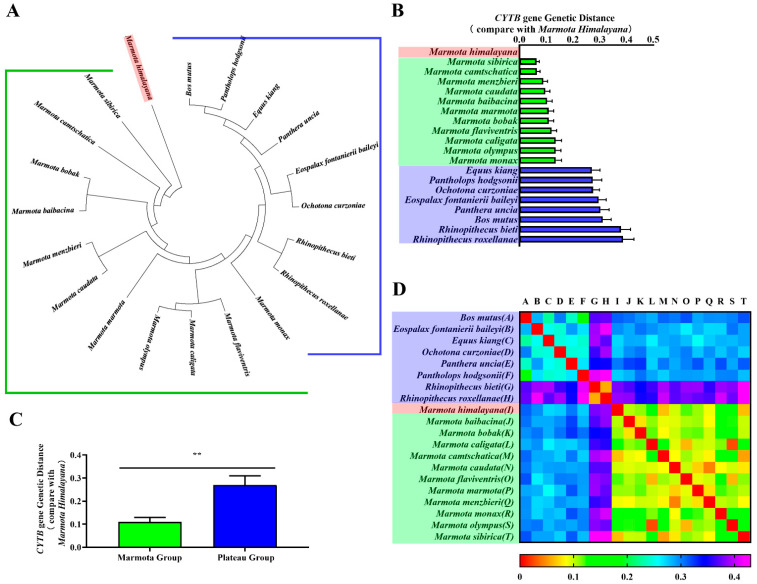
Phylogenetic analyses of *CYTB* gene. Phylogenetic trees for the *CYTB* gene were constructed in MEGA (**A**). The genetic distance from the Himalayan marmot to each member of the marmot group and the plateau group are represented as a bar chart (**B**). One-way ANOVA showed a significant difference (** *p* < 0.01) between the groups (**C**). A heatmap represents the correlation coefficients of the *CYTB* gene genetic distance for all species (**D**).

**Figure 3 animals-11-00251-f003:**
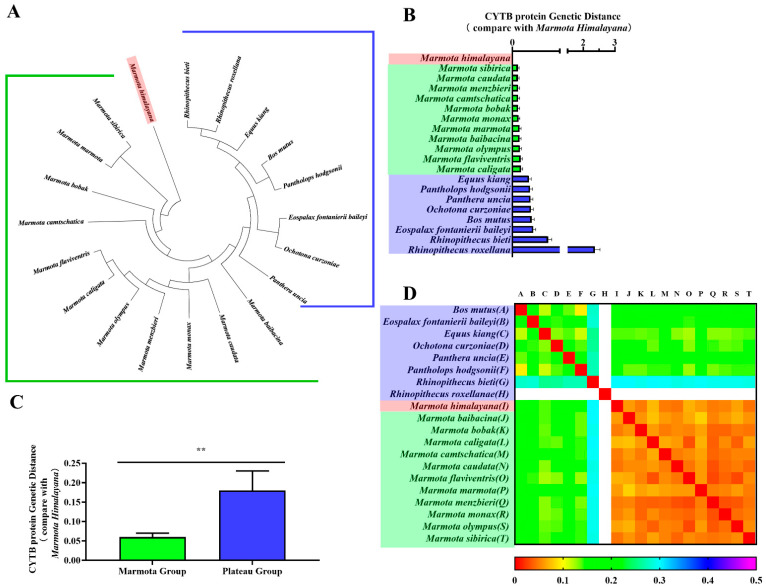
Phylogenetic analyses of CYTB protein. Phylogenetic trees for the CYTB protein were constructed in MEGA (**A**). The genetic distance from the Himalayan marmot to each member of the marmot group and the plateau group are represented as a bar chart (**B**). One-way ANOVA showed a significant difference (** *p* < 0.01) between the groups (**C**). A heatmap represents the correlation coefficients of the CYTB protein genetic distance for all species (**D**).

**Figure 4 animals-11-00251-f004:**
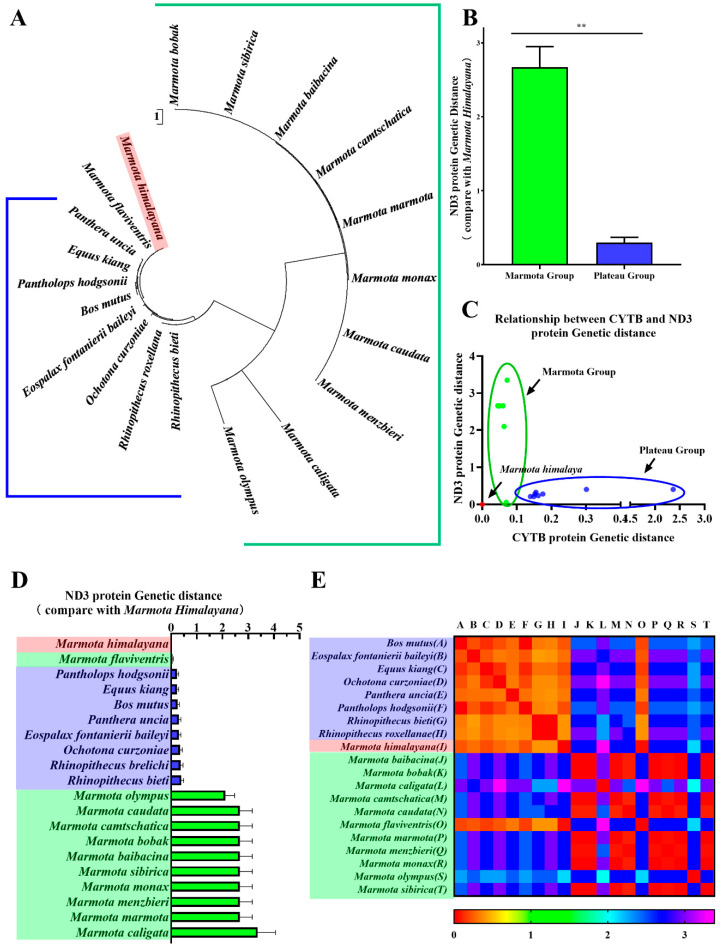
Phylogenetic analyses of ND3 protein. Phylogenetic trees of ND3 protein were constructed in MEGA with the scale bar set to 1 to measure genetic distances (**A**). One-way ANOVA revealed a significant difference between the groups (** *p* < 0.01) (**B**). A scatter plot shows the correlation between genetic distances calculated for CYTB and ND3 amino acid sequences by species (**C**). Genetic distances of the marmot group and plateau group were plotted as a bar chart (**D**). A heatmap represents the correlation coefficients of ND3 protein genetic distance for all species (**E**).

**Figure 5 animals-11-00251-f005:**
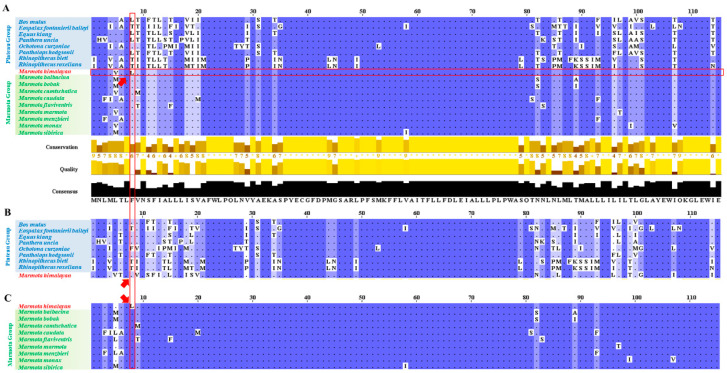
Nonconserved residues of ND3 amino acid sequence. The nonconserved ND3 amino acid sequence for all species (**A**), the plateau group (**B**), and the marmot group (**C**) were aligned in Jalview. The capital letters follow the standard notation for amino acids. Dark blue areas are highly conserved, while the light blue and white points represent divergent sequences. The nonconserved amino acid site in the Himalayan marmot is marked by a red rectangle, with the amino acid for the Himalayan marmot itself marked by red arrows.

**Figure 6 animals-11-00251-f006:**
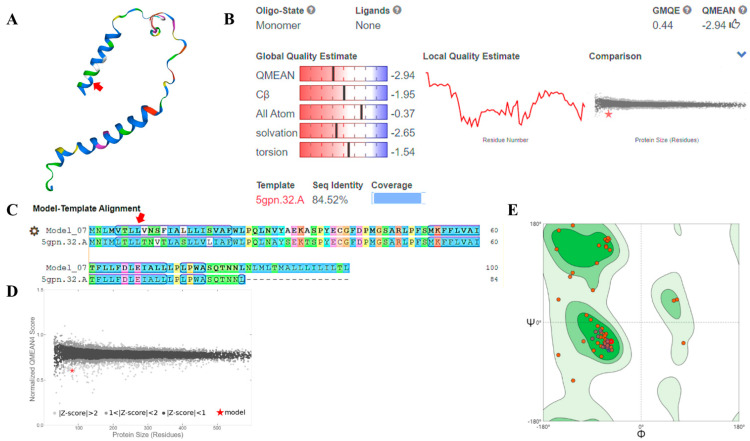
Modelling results and evaluations of ND3 protein tertiary structure of Himalayan marmot. Using the Swiss-Model website, we obtained the ND3 protein tertiary structure for the Himalayan marmot (**A**). The accuracy of the target protein structure is evaluated using QMEAN, sequence identity (**B**), target-template sequence alignment (**C**), Z-score (**D**), and Ramachandran plots (**E**). ψ and φ are dihedral angle of acid residues. The eighth amino acid in the sequence is marked with a red arrow, and the red star means the location of 5gqpn.32.A (model) in Z-score.

**Table 1 animals-11-00251-t001:** The Species and Accession Number of Cyt B and ND3.

Species	Common Name	Elevation (meter)	Classification		Accession Number			
			Marmot	Plateau	Cyt B Protein	*Cyt B* Gene	ND3 Protein	*ND3* Gene
*Bos mutus*	*wild yak*	3000–6000	No	Yes	AAX53006.1	KM280688.1	AKG95369.1	Unpublish
*Eospalax fontanierii* *baileyi*	*plateau zokor*	2800–4300	No	Yes	ADU18101.1	GU339023.1	YP_006493403.1	Unpublish
*Equus kiang*	*kiang*	3600–5400	No	Yes	AEU10871.1	JF718885.1	AFQ41487.1	Unpublish
*Ochotona curzoniae*	*plateau pika*	3100–5100	No	Yes	AAG00197.1	KM225712.1	ABP99054.1	Unpublish
*Panthera uncia*	*Uncia uncia*	2500–5000	No	Yes	AKG95247.1	KF990331.1	ABP73333.1	Unpublish
*Pantholops hodgsonii*	*Tibetan antelope*	3700–5500	No	Yes	AAC31679.1	AF034724.1	YP_337831.1	Unpublish
*Rhinopithecus bieti*	*Black snub-nosed* *monkey*	2500–5000	No	Yes	ADI32831.1	AY232663.1	AAD08821.1	Unpublish
*Rhinopithecus roxellana*	*Golden snub-nosed* *monkey*	1500–3300	No	Yes	AIX11529.1	AF262360.1	AAD04665.1	Unpublish
*Marmota himalayan*	*Himalayan marmot*	2900–5500	Yes	Yes	ACT78394.1	GQ329722.1	YP_006576328.1	JF313281.1
*Marmota baibacina*	*Altai marmot*	<4000	Yes	No	AAD45205.1	AF100714.1	AEL87735.1	JF313282.1
*Marmota bobak*	*bobak marmot*	No data	Yes	No	AAD45203.1	AF143917.1	AEL87708.1	JF313273.1
*Marmota caligata*	*Hoary marmot*	>0	Yes	No	AAD45209.1	KJ458055.1	AEL87714.1	JF313275.1
*Marmota camtschatica*	*Black-capped marmot*	<1900	Yes	No	AAD45206.1	AF100715.1	AEL87717.1	JF313276.1
*Marmota caudata*	*Long-tailed Marmot*	3200–3800	Yes	No	AAD29732.1	AF100716.1	AEL87726.1	JF313279.1
*Marmota flaviventris*	*Yellow-bellied marmot*	>2000	Yes	No	AAD29733.1	AF143927.1	YP_009632421.1	JF313280.1
*Marmota marmota*	*Alpine marmot*	600–3200	Yes	No	AAD29736.1	AF143930.1	AEL87741.1	JF313284.1
*Marmota menzbieri*	*Menzbier’s marmot*	2000–3400	Yes	No	AAD29738.1	AF143931.1	AEL87744.1	JF313285.1
*Marmota monax*	*Woodchuck*	No data	Yes	No	AAD45210.1	AF100719.1	AEL87747.1	JF313286.1
*Marmota olympus*	*Olympic marmot*	>1990	Yes	No	AEL87706.1	AF111182.1	AEL87752.1	JF313288.1
*Marmota sibirica*	*Mongolian marmot*	0–3800	Yes	No	AAD29745.1	AF143937.1	AEL87758.1	JF313289.1

## Data Availability

The raw/processed data required to reproduce these findings cannot be shared at this time as the data also forms part of an ongoing study.

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
