# Peer review of "Convergent Evolution of Himalayan Marmot with Some High-Altitude Animals through ND3 Protein"

_animals, 2021, doi:10.3390/ani11020251_

Round 1

Reviewer 1 Report

The study of Bao et al. represents a clear example of convergent evolution on genetic adaptations to high altitudes among several mammal species. The study is well designed and the overall structure of the manuscript is clear. 

There are several minor reformatting issues that I highlighted in the attached pdf file, and some parts the extensive English editing is needed.

Reviewer 2 Report

The manuscript describes analysis of the Himalayan marmot to other marmots, and other animals living on the Qinghai-Tibet Plateau using phylogenetic analysis of two genes/proteins in each animal studied. 

Some English language editing will improve sentence structure.  For example, from the abstract “This could lead to different protein spaces and functional changes; for instance, the structure and sequence similarity were 84.52% with pigs (Sus scrofa).” Long sentence, and the phrase "protein spaces" is not clear.

The methods fail to convey the way in which the authors obtained standard deviations from phylogenetic tree analysis (Figure 2B, 3B), nor the methods for statistical analysis of groupings (Figure 2C, 3C). More detail is required as to how a range of distances were obtained for each species (mean +/- standard deviation), and how these were grouped to great Figure 2C, 3C, as these are not necessarily standard practices. 

The final sentence of the abstract is too strong for the data shown and “have been adapted” should be changed to “suggests they have been adapted”, as there has not be a direct demonstration.

Reviewer 3 Report

In this paper, Authors compare the Himalayan marmot with other marmots and with different species living in cold environmental conditions. By using CYTB sequences (DNA and protein) they observe, as expected, a lower distance between marmots group. By using ND3 gene data, the result is reversed. As a consequence, Himalayan marmot underwent convergent evolution with species living in cold conditions.

I would like to address some questions:

  • As I understand it, one amino acid mutation moves the Himalayan marmot from one group to the other. Since we do have a little experience with similarity analyses at the level of both DNA and protein level, may I ask to Authors to try to see what happens if they create a single amino acid mutation counterbalancing the one they observe. That is, a mutation bringing back the Himalayan marmot back to the group of marmots?
  • Analyses based on single sequences available in international databases suffer the usual obvious problem: population variability. Could we have polymorphism at the eight amino acid of the ND3 protein of the Himalayan marmot? And with which consequences?
  • “The properties of a protein are determined by its shape”. True! Could we see in figure 6 the three-dimensional structure of the ND3 protein of another marmot living in the plain and of one of the animals living in the plateau? Just to see whether differences appear.
  • The simple summary should be rewritten, according to me.
